# Immune correlates of early clearance of *Mycobacterium tuberculosis* among tuberculosis household contacts in Indonesia

Todia P. Setiabudiawan [1,2], Lika Apriani[2,3], Ayesha J. Verrall[4], Fitria Utami[2], Marion Schneider[5], Agnes R. Indrati[6], Pauline P. Halim[7], Paulina Kaplonek[8], Hadar Malca[8], Jessica Shih-Lu Lee[8], Simone J. C. F. M. Moorlag[1], L. Charlotte J. de Bree[1], Vera P. Mourits[1], Leo A. B. Joosten [1,9], Mihai G. Netea [1,10], Bachti Alisjahbana[2,11], Ryan P. McNamara[8,12], Galit Alter [8], Arjan van Laarhoven [1], James E. Ussher [5], Katrina Sharples[13], Valerie A. C. M. Koeken [1,14], Philip C. Hill[15] & Reinout van Crevel [1,16] ✉

Some individuals, even when heavily exposed to an infectious tuberculosis patient, do not develop a specific T-cell response as measured by interferon-gamma release assay (IGRA). This could be explained by an IFN-γ-independent adaptive immune response, or an effective innate host response clearing *Mycobacterium tuberculosis (Mtb)* without adaptive immunity. In heavily exposed Indonesian tuberculosis household contacts (n = 1347), a persistently IGRA negative status was associated with presence of a BCG scar, and - especially among those with a BCG scar - with altered innate immune cells dynamics, higher heterologous (*Escherichia coli*-induced) proinflammatory cytokine production, and higher inflammatory proteins in the IGRA mitogen tube. Neither circulating concentrations of *Mtb*-specific antibodies nor functional antibody activity associated with IGRA status at baseline or follow-up. In a cohort of adults in a low tuberculosis incidence setting, BCG vaccination induced heterologous innate cytokine production, but only marginally affected *Mtb*-specific antibody profiles. Our findings suggest that a more efficient host innate immune response, rather than a humoral response, mediates early clearance of *Mtb*. The protective effect of BCG vaccination against *Mtb* infection may be linked to innate immune priming, also termed 'trained immunity'.

Some people who are heavily exposed to an infectious tuberculosis patient do not develop evidence of an antigen-specific T-cell response, as measured with an interferon-gamma release assay (IGRA). We have previously found that approximately one-quarter of heavily exposed tuberculosis household contacts in Indonesia do not develop a positive IGRA during three months follow-up[1]. One might argue that these

individuals either clear inhaled *Mycobacterium tuberculosis (Mtb)* through a protective innate host response, or that they develop an interferon-γ (IFNγ) independent adaptive immune response.

Interestingly, tuberculosis household contacts with a BCG scar showed a ~50% lower risk of IGRA conversion compared to unvaccinated individuals[1]. Protection associated with BCG scars decreased

with increasing *Mtb* exposure and correlated with the heterologous innate immune response[2]. It should be noted that not all individuals develop a scar after BCG vaccination[3]. Still, these data suggest that BCG-induced innate immune priming (also termed 'trained immunity'), which has been shown to protect against *Mtb* in experimental models[4–6], may clear inhaled *Mtb* before an adaptive immune response (as measured with an IGRA) can develop.

Rather than reflecting protective innate immune clearance, a persistently negative IGRA status among heavily and recently exposed household contacts might also be explained by an IFN-γ-independent adaptive immune response. In Uganda, contacts who had tested IGRA- and tuberculin skin test (TST)-negative over several years (so-called 'resisters'), had detectable IFN-γ-negative T-cell responses to ESAT6/CFP10, the antigens used for IGRA-testing and absent in BCG[7]. They also had similar concentrations of IgG, IgM, and IgA antibodies to different *Mtb* antigens as IGRA-positive contacts[7]. Other studies, in humans[8] as well as primates[9], have also found anti-*Mtb* antibodies and suggested that they may protect against *Mtb* infection as well as TB disease in an IFN-γ-independent way[10].

To improve our understanding of the correlates of protection against *Mtb* infection, we examined innate immune cell phenotype and function, and a broad range of anti-*Mtb* specific antibody features in heavily exposed tuberculosis household contacts in Indonesia, as well as in BCG-vaccinated adults in a low-TB incidence setting.

## Results

### Characteristics of tuberculosis household contacts in Indonesia

Among 1347 heavily exposed tuberculosis household contacts, after the exclusion of individuals with active TB, 780 (57.9%) had a positive, and 433 (32.1%) had a negative IGRA result at baseline. The median age of contacts was 31 (IQR: 17 - 47) for IGRA-positive and 22 (IQR: 12-39) years for IGRA-negative individuals, with 10.5% and 15.0% of subjects below 10 years of age, respectively. Baseline IGRA-positive individuals had spent more time with the index patient, and more often slept in the same room with them (Table 1). Among household contacts with a negative IGRA at baseline, 116 (26%) converted to a positive IGRA at 14 weeks. IGRA conversion was associated with higher exposure, while a persistently IGRA negative status was associated with the presence of a BCG scar (Supplementary Table S1). This association had an interaction with the level of exposure and with age, with lower protection from BCG against IGRA conversion among household contacts with higher exposure or older age. However, the relation of BCG with IGRA conversion remained significant in multivariate analysis (aRR 0.56 [95% CI, 0.40–0.77]; *P* < 0.001). To strengthen the phenotypes, a strict cut-off value was used for negative IGRA results (<0.15 IU/mL) and conversion to a positive IGRA result at 14 weeks (> 0.7 IU/mL). Using these stricter criteria, we compared 51 participants classified as IGRA converters and 237 as persistently IGRA-negative individuals (Supplementary Fig. S1 and Supplementary Table S1). Using these IGRA cut-offs, differences between IGRA converters and persistently IGRA-negative individuals in the level of exposure to the index patient, and in the proportion of individuals with a BCG scar (RR 0.35 [95%CI, 0.21 - 0.58]; *P* < 0.001, Supplementary Table S1) were more pronounced. Also, IGRA conversion was more frequent among HHCs of index patients with Mtb L2 (Beijing) genotype strains isolated from sputum compared to those infected with other genotype strains, and BCG vaccination appeared less protective against infection by L2 strains[11]. Using stricter IGRA criteria, we saw a stronger relative risk (RR) for infection after exposure to L2 versus other genotype strains (RR 1.84 [95% CI, 1.11-2.97], *P* = 0.015 with strict criteria vs RR 1.44 [95% CI, 0.98-2.10], *P* < 0.001 with the manufacturer IGRA criteria, Supplementary Table S2A). Similarly, the genotype-dependent difference in protection conferred by BCG vaccination was stronger with stricter IGRA cut-offs (Supplementary Table S2B).

### Different dynamics of innate immune cells in IGRA negative contacts

Among a subset of household contacts with a negative IGRA at baseline that had given informed consent for an additional blood draw at week 2 and week 14 (*N* = 102), 16 different innate immune cell subsets were measured using flow cytometry. For further analysis, we included participants who had data for both time points, including 22 IGRA converters and 48 persistently IGRA-negative individuals. At week 2, there were no statistically significant differences in innate immune cell numbers between groups (Supplementary Fig. S2). When results at week 2 and week 14 were compared, innate immune cell numbers showed no statistically significant change in IGRA converters, while persistently IGRA-negative individuals showed a significant reduction in the numbers of CD14^hiCD16^- classical monocytes, CD14^hiCD16^+ intermediate monocytes, CD14^lowCD16^+ non-classical monocytes, CD16^+ mature granulocytes, CD16^dim immature granulocytes, and Vδ2^- γδ T cells (Fig. 1B). When the analysis was restricted to persistently IGRA-negative contacts, the decrease in numbers of total monocytes, classical monocytes, intermediate monocytes, non-classical monocytes, mature granulocytes, and Vδ2^- γδ T cells was more pronounced among individuals with a BCG scar (*N* = 38) compared to those without (*N* = 10) (Fig. 1C), while this subgroup of persistently IGRA-negative individuals with a BCG scar also showed a significant reduction in CD56^dim NK cells (Fig. 1C).

### Association of innate cytokine production with IGRA status

We next examined how innate immune markers correlated with IGRA status (Fig. 2A). First, we compared baseline production of TNF, IL-8, IL-6, IL-1β, IL-1Ra, and IL-10 upon stimulation with *Mtb*, BCG, and *E. coli* as a heterologous stimulus. As expected, baseline IGRA-positive individuals (*N* = 145) showed higher cytokine production upon *Mtb* and BCG stimulation compared to baseline IGRA-negative individuals (*N* = 328) (Fig. 2B, C). Also, logistic regression showed a strong association of innate cytokine production after both *Mtb* and BCG stimulation with IGRA positivity at baseline. (Fig. 2D). Among baseline IGRA-negative individuals, those who remained IGRA-negative after 14 weeks (*N* = 237) showed higher innate cytokine production upon *E. coli* stimulation compared to those whose IGRA converted to positive (*N* = 91) (Fig. 2B, C), and logistic regression showed IL-6 and IL-8 production upon *E. coli* stimulation to be associated with persistently IGRA-negativity at follow-up (Fig. 2E). Interestingly, the association of *E.coli*-induced production and persistently IGRA-negativity at follow-up was stronger in contacts with a BCG scar compared to those without for IL-8, TNF and IL-6 (Fig. 2F).

### Associations of baseline IGRA supernatant inflammatory proteins with IGRA status at follow-up

Building on the ex vivo cytokine production data, we then measured inflammatory proteins in supernatants of baseline IGRA nil and mitogen tubes. Several proinflammatory proteins (ADA, MCP-3 [CCL7], TWEAK, IL-17C, and IL-18) showed significantly higher concentrations (logistic regression with adjustment for age, sex, BMI, and exposure risk score) in baseline IGRA supernatants of contacts whose IGRA remained negative compared to those whose IGRA converted to positive at 14 weeks (Fig. 3A). Differentially abundant proteins showed consistent results in nil and mitogen tubes (Fig. 3B). Besides the aforementioned proteins, five additional inflammatory proteins in mitogen-stimulated IGRA supernatants (CSF-1, CD244, DNER, CD6, and VEGFA) correlated with IFN-γ (TBAg − Nil) levels at 14 weeks after adjustment for age, sex, BMI, and exposure risk score (Fig. 3D).

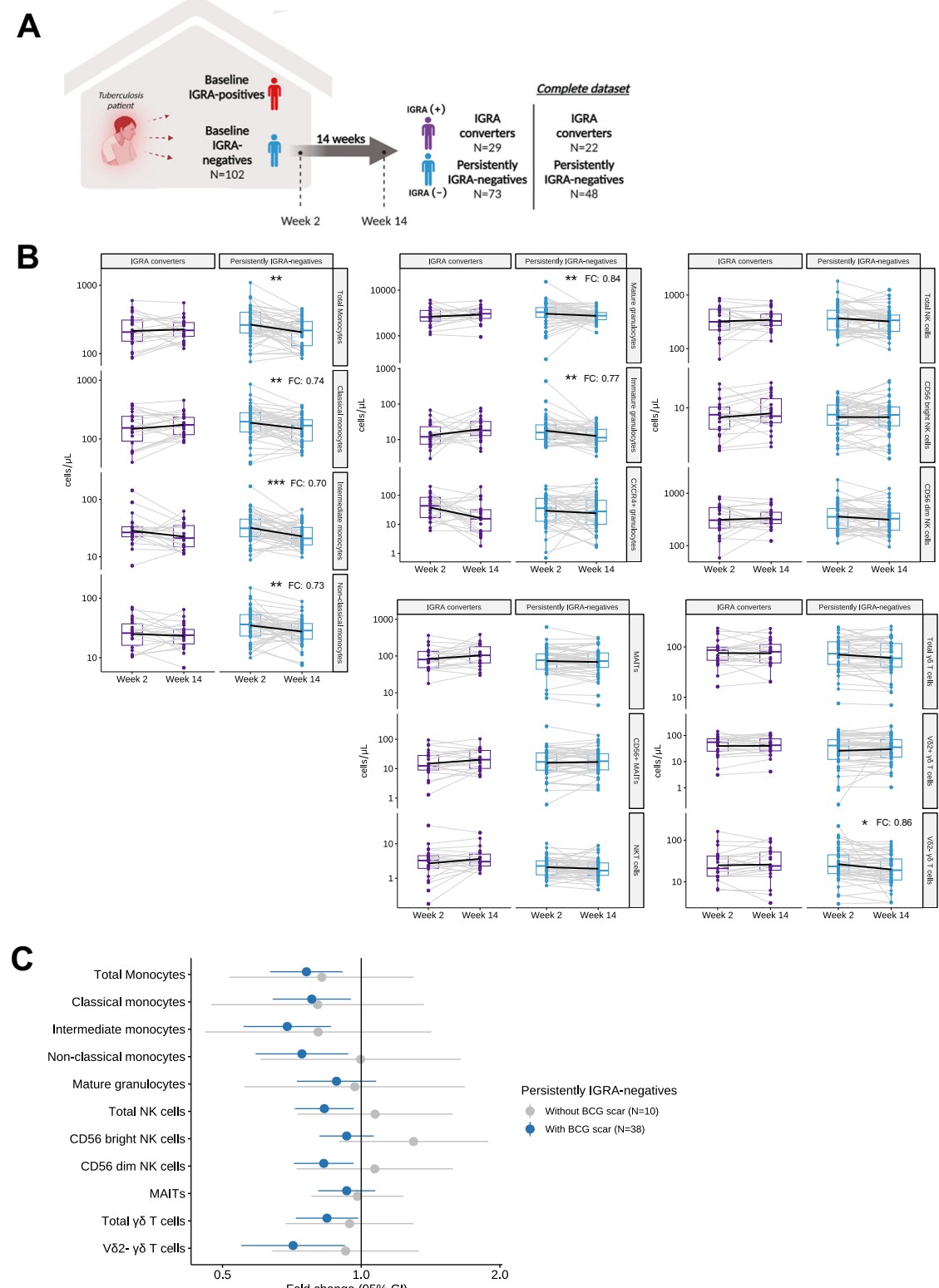

**Fig. 1 | The dynamics of innate immune cells in IGRA-converters and persistently IGRA-negative individuals. A** Overview of the flow cytometry dataset. Created in BioRender. Setiabudiawan, T. (2025) https://BioRender.com/w61f180. **B** Frequencies of circulating innate immune cells (numbers / μL blood) were compared between week 2 and week 14 in IGRA converters (*N* = 22), and persistently IGRA-negative individuals (*N* = 48, [FDR <0.1, <0.05, <0.01; \*, \*\*, \*\*\*; FC = median fold change]). **C** Persistently IGRA-negatives with BCG scar (*N* = 38) showed a larger decrease in cell numbers than participants without BCG scar (*N* = 10) in the innate circulating immune cells from week 2 to week 14.

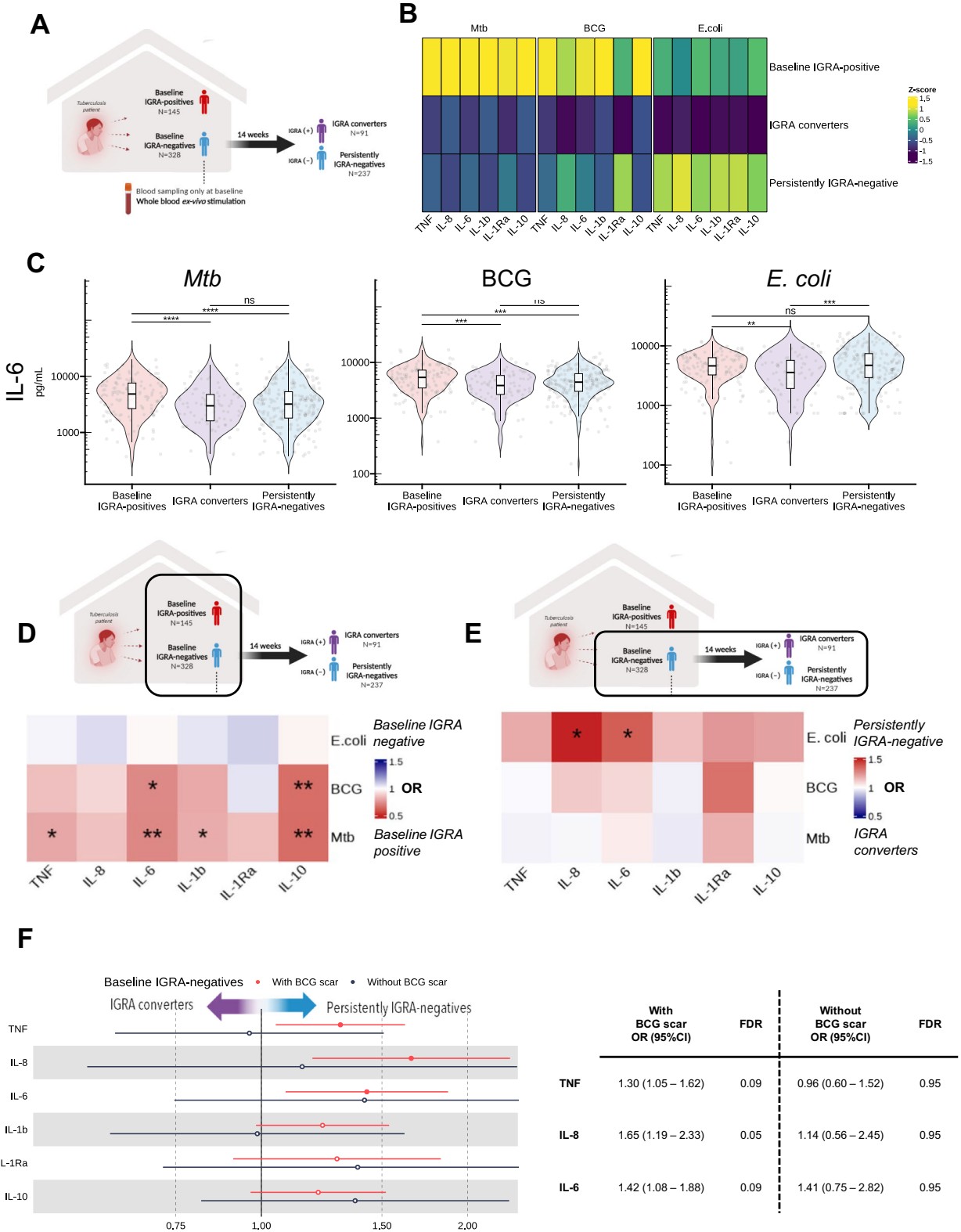

## Antibodies and antibody function in relation to IGRA status

Antibodies were measured at baseline in randomly selected IGRA-positive ($n = 100$) and all IGRA-negative contacts ($N = 433$). Similar to the larger cohort, IGRA-positive individuals more often slept in the same room as the index case, spent more hours in contact with them, and had a higher likelihood of living with an index case with cavitary disease on chest X-ray (Supplementary Table S3). After filtering for antibodies with a ratio higher than those measured in PBS, 25 out of 55 *Mtb*-antigen-specific antibody isotypes were selected for analysis (Supplementary Fig. S4). Antibodies showed a moderate association with age, sex, and BMI (Supplementary Fig. S5A). No antibodies measured at baseline were significantly different between IGRA-positive and IGRA-negative individuals (Fig. 4A). Partial least squares – discriminant analysis (PLS-DA) showed overlapping clusters of IGRA-

**Fig. 2 | Study outline and ex vivo cytokine production. A** Baseline whole blood ex vivo cytokine production, compared between baseline IGRA-negative ($N = 328$) and IGRA-positive individuals ($N = 145$), and between IGRA converters ($N = 91$) and persistently IGRA-negative individuals ($N = 237$). Created in BioRender. Setiabudiawan, T. (2025) https://BioRender.com/k78q817. **B** Cytokine production following stimulation with *Mtb*, BCG, and *E.coli*, with higher *Mtb*-induced cytokine production in baseline IGRA-positive individuals, and higher *E. coli*-induced production in persistently IGRA-negative individuals. **C** *Mtb*, BCG, and *E.coli*-induced IL-6 production (as a representative), stratified for IGRA-status (Mann-Whitney *U* test

after correction for multiple testing). Association between cytokine production and IGRA status at baseline (**D**) and 14 weeks (**E**), expressed as odds ratio (OR) using logistic regression adjusting for age, sex, BMI, exposure risk score, blood monocyte count, blood lymphocyte count, and batch. **F** Relation between baseline ex vivo cytokine *E.coli*-induced production (in IGRA-negative individuals) and IGRA status at 14 weeks, shown as odds ratios, stratified for BCG vaccination status. All models were corrected for multiple testing (Benjamini-Hochberg). (FDR <0.1, <0.05, <0.01, <0.001; closed circle & *, **, ***, ****).

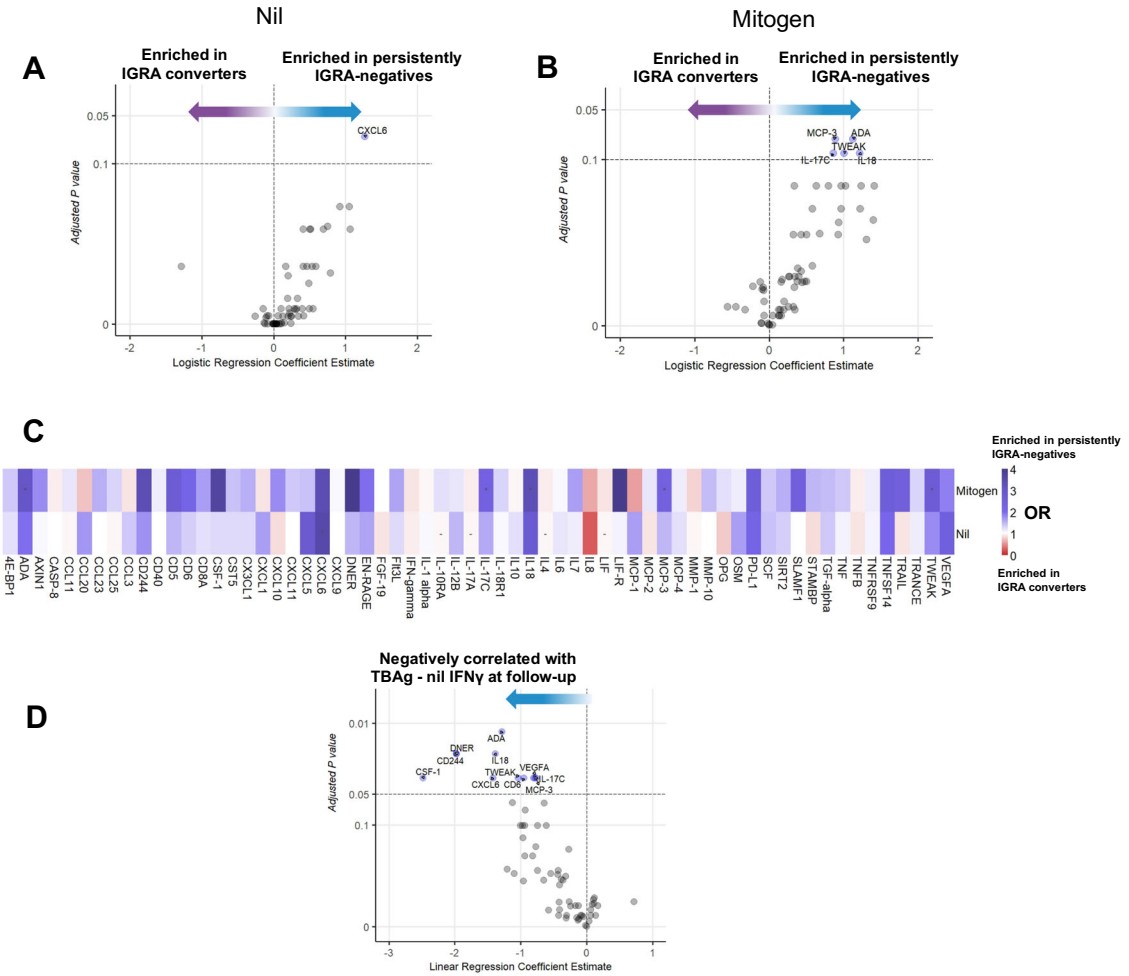

**Fig. 3 | Inflammatory proteins in IGRA supernatants.** Inflammatory proteins relative concentrations (NPX unit, log$_2$ transformed) in IGRA supernatants (nil and mitogen) at baseline were compared between IGRA converters ($N = 48$) and persistently IGRA-negative individuals ($N = 128$). **A** A persistently IGRA-negative status was associated with higher CXCL6 in the baseline IGRA nil tube, and (**B**) with ADA, MCP-3 (CCL7), TWEAK, IL-17C, and IL-18 in the baseline mitogen tube (logistic regression adjusted for age, sex, BMI, and exposure risk score; FDR <0.1).

**C** Associations between persistently IGRA-negative status and concentrations of all proteins measured in IGRA nil and mitogen tube (Odds ratios [OR], adjusted for age, sex, BMI, and exposure risk score). **D** In the mitogen tubes, the same proteins, as well as CSF-1, DNER, CD244, and VEGFA, showed a correlation with quantitative TBAg - nil IFNγ IGRA results after correction for multiple testing with a lower FDR cutoff of 0.05.

positive and IGRA-negative individuals (Fig. 4B). Also, no antibody levels were associated with IGRA status at baseline based on logistic regression analysis adjusting for age, sex, and BMI, and correction for multiple testing (Fig. 4C).

We next examined if antibodies against *Mtb* measured at baseline were associated with the risk of IGRA-conversion, using strict IGRA cut-off criteria. No antibodies were significantly different between persistently IGRA-negative individuals ($N = 237$) and IGRA converters ($N = 51$; Fig. 4D). PLS-DA showed no differences between the groups (Fig. 4E). In addition, no antibodies were associated with the risk of IGRA conversion in logistic regression (Fig. 4F). Moreover, when analysis was

limited to household contacts with a BCG-scar, no differences between groups were found in antibody concentrations (data not shown).

Antibodies can exert their function through lysis of infected cells by complement activation, or promote cellular or neutrophil phagocytosis, which might add to clearance of *Mtb* upon exposure. Focusing on LAM-specific antibodies which had the highest variable of importance projection scores in the PLS-DA (Supplementary Fig. S6), we examined if antibody-dependent complement deposition (ADCD), antibody-dependent cellular phagocytosis (ADCP), and antibody-dependent neutrophil phagocytosis (ADNP) were associated with IGRA conversion. Using our stricter IGRA criteria and a subset of

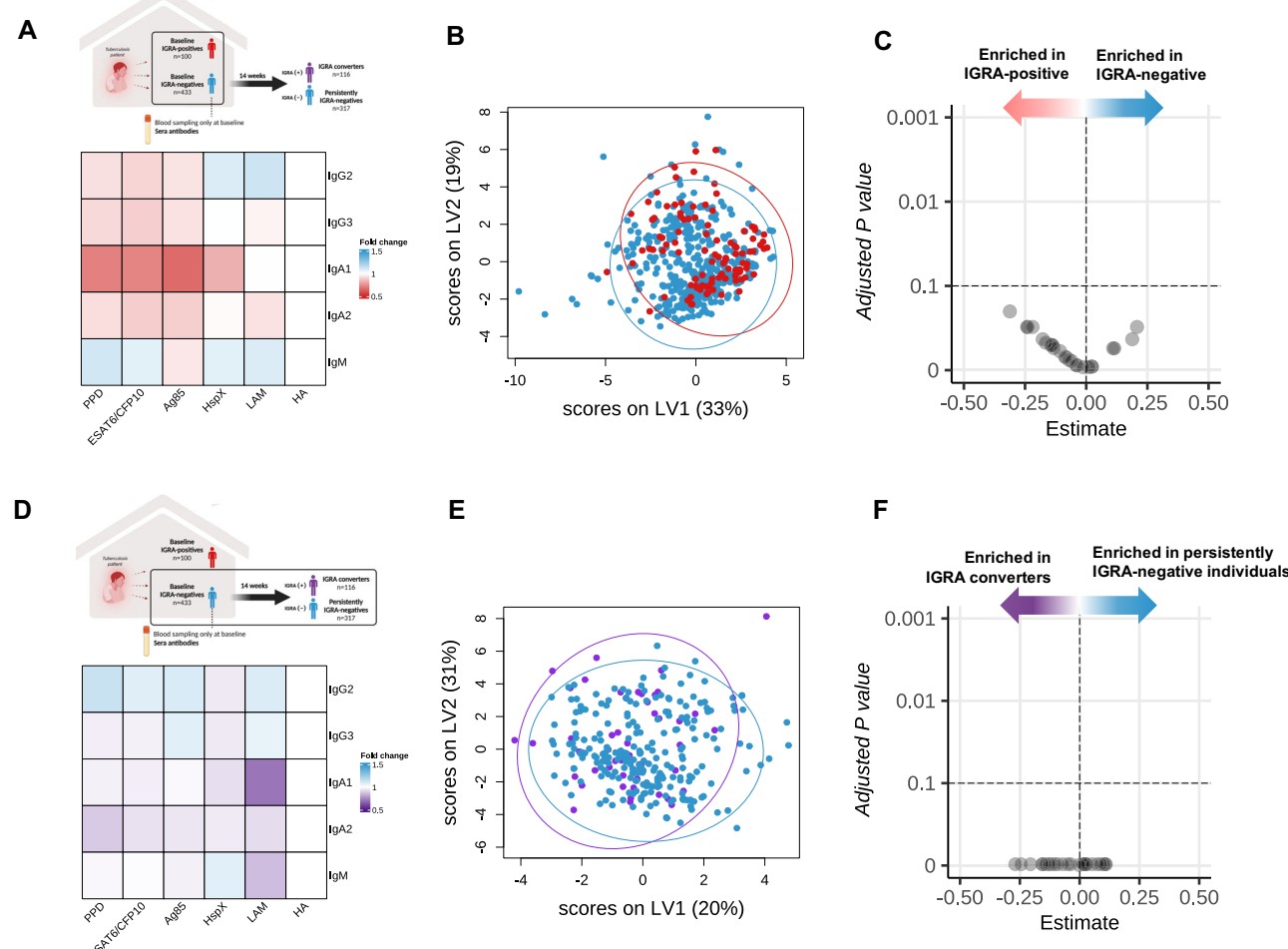

**Fig. 4 | Antibody profiles according to IGRA-status at baseline and follow-up.** Antibody profiles were compared between IGRA-positive ($N = 100$) and IGRA-negative ($N = 433$) tuberculosis household contacts; and between IGRA converters ($N = 51$) and persistently IGRA-negative individuals ($N = 237$), based on strict IGRA criteria (<0.15 IU/mL for negative and > 0.70 IU/mL for positive). Fold differences in antibody levels (shown as the ratio of antibodies corrected for the positive control hemagglutinin [HA]), are shown according to IGRA status at baseline (**A** red: higher antibody levels in IGRA-positive individuals) and follow-up (**D** purple: higher antibody levels in IGRA converters). No difference reached statistical significance, thus, numbers not shown in the heatmap (Mann-Whitney U test; FDR <0.1). **B** Partial least squares discriminant analysis (PLS-DA) using the selected 25 antibodies was used to visualize differences in antibody levels between baseline IGRA-positive (red) and -negative (blue) individuals, and (**E**) between IGRA converters (purple) and persistently IGRA-negative individuals (blue). **C** In logistic regression, no antibody was associated with IGRA status at baseline or follow-up (**F**) after adjustment for sex, age, BMI, and exposure risk score.

individuals matched for age and sex, IGRA converters ($N = 50$) had higher MFI for LAM-dependent ADCD than persistently IGRA-negative individuals ($N = 50$), while ADCP and ADNP showed no difference based on univariate testing (Supplementary Fig. S7A). However, in logistic regression adjusting for age, sex, BMI, and exposure risk score there was no association between ADCD, ADCP, or ADNP with IGRA status during follow-up (Supplementary Fig. S7B).

### Effect of BCG vaccination on cytokine production and anti-*Mtb* antibodies

To further investigate the induction of innate immune responses and antibody production after mycobacterial stimulation in vivo, we next used a cohort of healthy volunteers vaccinated with BCG in a low-TB incidence setting[12]. The presence of a BCG scar had shown strong relations with immune markers among individuals in a high-burden setting (Indonesia) who were examined years after they had been vaccinated at birth, and we went to a low-burden setting to examine this effect of BCG (before and three months after vaccination) in the absence of possible confounding by exposure to *M. tuberculosis*.

As expected, vaccination with BCG (an attenuated form of *M. bovis* which shares 99% similarity with *Mtb*)[13-15] led to an increase in ex vivo *Mtb*-induced IFN-γ production, but also to an increase in innate cytokines (Fig. 5A). As previously shown, BCG vaccination also led to increased heterologous cytokine production, although not in all individuals, as depicted for stimulation with *Staphylococcus aureus* in Fig. 5B. To examine if BCG vaccination induced anti-*Mtb* antibodies, we measured concentrations of 5 antibody isotypes and binding level of 2 Fc-receptors, to 9 *Mtb* antigens standardized to hemagglutinin. After 90 days, when corrected for multiple testing, several *Mtb*-specific IgG3 showed a statistically significant, albeit minimal increase, while several *Mtb*-specific IgM antibodies showed a minimal decrease (Fig. 5C, D and Supplementary Fig. S8). Differences in antibody concentration between day 0 and day 90 were not due to (seasonal) changes in anti-HA antibodies for which we normalized (Fig. 5D).

### Discussion

In a tuberculosis household study in Indonesia, approximately one-fourth of heavily exposed contacts still had a negative IGRA three

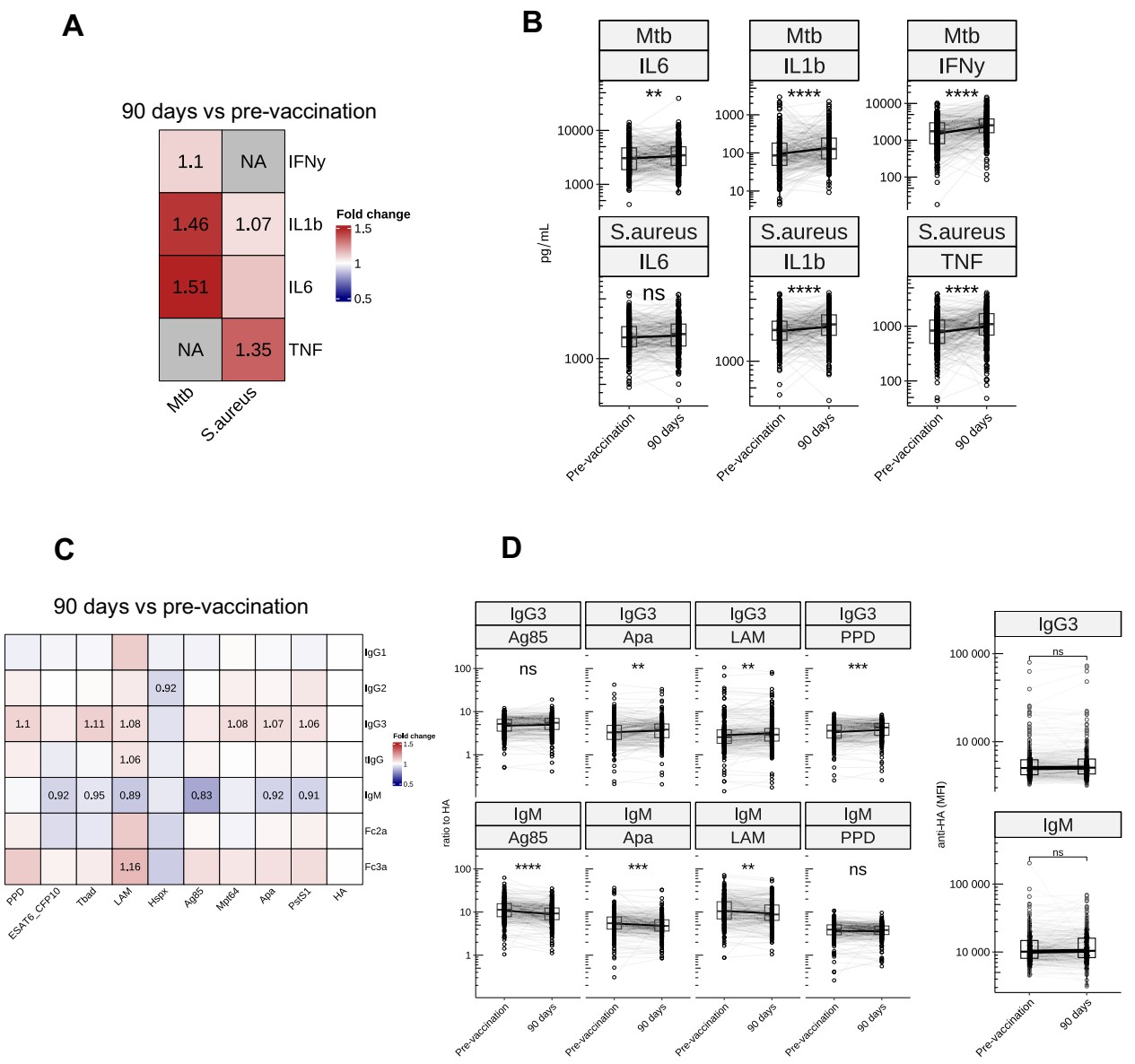

**Fig. 5 | Effect of BCG vaccination on cytokine production and *Mtb*-specific antibodies.** Heatmap showing fold change (**A**) and paired boxplot (**B**) of ex vivo cytokine production in Dutch healthy adults (*N* = 298) before and 90 days after BCG vaccination. Shown are 24 h stimulation of PBMCs with *Mtb* and *S. aureus*, and 7-day stimulation of *Mtb* for IFN-γ. **C** shows a fold change of hemagglutinin (HA)-standardized anti-*Mtb* antibody levels at day 90 compared to the pre-vaccination, with statistically significant fold changes shown in numbers. **D** Shows changes in IgG3 and IgM antibodies against different *Mtb* antigens, and raw anti-HA IgM and IgG3 MFI pre-vaccination and 90 days post-vaccination were used for normalization. FDR <0.1, <0.05, <0.01, <0.001; *, **, ***, ****.

months after tuberculosis diagnosis of the index case. Examining their innate immune response as a possible mechanism to remain uninfected, individuals with a persistently negative IGRA showed a stronger reduction of innate immune cells over time compared to IGRA converters, and higher heterologous production of cytokines and inflammatory proteins at baseline. No differences were found in baseline concentration or function of anti-*Mtb* antibodies, as a possible marker of an IFN-γ independent adaptive immune response. Among contacts with a BCG scar, which was associated with a persistently negative IGRA status, more pronounced differences were seen in innate immune cell numbers and function between IGRA converters and persistently IGRA-negative individuals. Furthermore, in a low-incidence setting, adult BCG vaccination induced heterologous cytokine production, but only led to marginal changes in anti-*Mtb* antibodies.

A T cell-mediated IFN-γ response is important, but not sufficient for protection against tuberculosis[16]. T-cell mediated IFN-γ responses against *Mtb* antigens are used for diagnosis of *Mtb* infection, with IGRAs[17]. T-cell immunity is crucial for protection against tuberculosis, as shown by the fact that among people with HIV, loss of CD4 T-cells correlates with the risk of tuberculosis[18]. In addition, rare genetic defects have demonstrated the crucial role of IFN-γ-signaling in mycobacterial infections[19]. Nevertheless, high IGRA IFN-γ production, as a mirror of T cell-mediated immunoreactivity against *Mtb*, increases rather than reduces an individual's likelihood of developing TB disease[20]. Also, *Mtb* seems to benefit from T cell recognition, as evidenced by the hyper-conserved T cell epitope sequences in the *Mtb* genome[21]. In addition, the MVA85A vaccine, which induces robust secretion of IFN-γ by CD4 + T cells, showed no protection against TB disease in clinical trials[22,23]. As such, these studies strongly argue that

**Table 1 | Characteristics of tuberculosis household contacts according to baseline IGRA-status**

| | Baseline IGRA-positive[a] N = 780 | Baseline IGRA-negative[a] N = 433 | P-value[b] |
|---|---|---|---|
| **Case contact characteristics** | | | |
| Age | 31 (17–47) | 22 (12–39) | <0.001 |
| Female sex | 58% | 53% | 0.089 |
| Presence of BCG scar | 78% | 84% | 0.013 |
| Current and previous smoking | 35% | 31% | 0.27 |
| BMI, kg/m$^2$ | 21.6 (18.0–25.4) | 20.2 (16.8–24.4) | 0.001 |
| Diabetes[c] | 3.2% | 3.9% | 0.41 |
| **Exposure to the index case** | | | |
| Sleeping in the same room as the index case | 30% | 20% | <0.001 |
| Waking hours spent with the index case a day before enrollment | 5 (2–10) | 4 (1– 8) | 0.001 |
| Index case highest smear grade | | | <0.001 |
| *Scanty* | 5.4% | 8.3% | |
| *1+* | 18% | 28% | |
| *2+* | 26% | 25% | |
| *3+* | 50% | 38% | |
| Presence of cavities on chest x-ray of index | 56% | 44% | <0.001 |
| Extent of x-ray abnormalities | 50 (25–71) | 40 (25–59) | <0.001 |
| *M. tuberculosis* L2 (Beijing) strain in the index case | 35% | 28% | 0.022 |
| **Blood count parameters at baseline** | | | |
| Hemoglobin g/dL | 13.70 (12.80–14.90) | 13.70 (12.80–15.00) | 0.56 |
| Platelets 1000/mm$^3$ | 298 (257–351) | 305 (258–360) | 0.17 |
| Leukocytes 1000/mm$^3$ | 7.50 (6.50–8.90) | 7.40 (6.20–8.60) | 0.077 |
| Lymphocytes 1000/μL | 2.60 (2.15–3.20) | 2.60 (2.12–3.07) | 0.20 |
| Neutrophiles 1000/μL | 4.12 (3.35–5.07) | 4.03 (3.20–5.02) | 0.23 |
| Monocytes 1000/μL | 0.46 (0.35–0.60) | 0.43 (0.32–0.56) | 0.003 |
| **Quantitative IFNγ release assay result** | | | |
| IFNγ Nil tube IU/L | 0.15 (0.09–0.29) | 0.14 (0.08–0.28) | 0.042 |
| IFNγ TB-Nil tube IU/L | 2.8 (1.1–6.7) | 0.0 (0.0 – 0.1) | <0.001 |
| IFNγ Mitogen-Nil tube IU/L | 9.32 (3.72–10.00) | 8.68 (3.41–10.00) | 0.53 |

*BCG* Bacillus Calmette-Guerin, *BMI* body mass index, *IQR* interquartile range.
[a]Median (IQR); %.
[b]Mann-Whitney U test; Pearson's Chi-squared test; Fisher's exact test.
[c]Diabetes defined as follows: no diabetes, random capillary blood glucose > 101 mg/dL or hemoglobin A1c (HbA1c) <5.7%; prediabetes, HbA1c 5.7%–6.4%; diabetes, HbA1c ≥ 6.5.

innate or other CD4/IFN-γ-independent mechanisms are also required for protection against tuberculosis. It should be noted that the correlates of protection against *Mtb* infection and TB disease are not necessarily the same.

Determining why some individuals do not develop a positive T cell-dependent TST or IGRA despite heavy exposure to *Mtb* can help identify novel correlates of protection against *Mtb* infection. The terms 'early clearance'[24] and 'resisters' have been used to label this clinical phenotype[25]. We studied early clearance in tuberculosis contacts in the context of a well-defined exposure within a household, with a relatively short follow-up, while so-called resisters are tuberculosis contacts with negative TSTs and IGRAs despite living in a high-incidence setting for years. Early clearance can be defined as a relative, or dynamic, measure of protection against *Mtb* infection[26], as we and others have shown that it is less common with heavier *Mtb* exposure[1], or exposure to more virulent L2 (Beijing) genotype strains[11]. In contrast, resisters can be

seen as individuals who do not establish *Mtb* infection despite repeated tuberculosis exposure of varying intensity over a long period of time[25].

Our study on early clearance in tuberculosis household contacts in Indonesia points to a significant role of innate immunity in the early protective response against *Mtb*. This hypothesis is supported by the elevated heterologous production of proinflammatory cytokines and inflammatory proteins, both produced mainly by innate immune cells, in persistently IGRA-negative individuals. In addition, the reduction in innate cell numbers which was found among contacts with a repeatedly negative IGRA at follow-up likely reflects the resolution of a protective innate inflammatory resolution after early clearance of *Mtb*, similar to the decreasing monocyte to lymphocyte ratio which has been reported during treatment of tuberculosis patients[27] and after TB preventive therapy of *Mtb* infected individuals[28].

The different innate immune cell numbers and functions in 'early clearers' in our study likely reflect a trained immunity[26] endotype associated with rapid elimination of the mycobacteria. This is further supported by the observation that the differences in innate immune cell phenotype and heterologous cytokine production between IGRA converters and persistently IGRA-negative individuals were more pronounced when analysis was restricted to individuals with a BCG scar. These findings mimic those of studies focusing on BCG-induced trained immunity in tuberculosis. In mice, BCG vaccination induces trained immunity in hematopoietic stem cells, which upon adoptive transfer conferred protection against *Mtb* in non-vaccinated mice[4]. Similarly, in a macaque model with repeated limiting doses of *Mtb* challenge, pulmonary mucosal BCG vaccination induced a stronger trained immunity response[5] and longer delay of IGRA-conversion compared to intradermal BCG[6]. In mice, induction of trained immunity through beta-glucan administration also protected against *Mtb*[29]. Collectively, this suggests that induction of trained immunity may protect tuberculosis contacts against *Mtb* infection and might help the development of other interventions to prevent tuberculosis. New vaccines preferably should strengthen innate immune protection that can withstand intense *Mtb* exposure.

There is renewed interest in the possible protective role of antibodies against tuberculosis. In one study, compared to tuberculosis patients, individuals with latent *Mtb* infection showed a higher abundance, higher Fc receptor binding, and higher antibody-dependent cellular cytotoxicity for several *Mtb*-specific antibodies[30]. In another study, circulating anti-*Mtb* antibodies that conferred protection against tuberculosis in mice were found in a proportion of healthcare workers, but not in tuberculosis patients[31]. Also, 40 tuberculosis household contacts in Uganda who had remained TST and IGRA-negative for several years (so-called 'resisters') were found to have detectable levels of *Mtb*-specific antibodies, similar to 39 *Mtb* IGRA/TST-positive individuals[7]. In a study in South Africa, 30 TST/IGRA-negative miners showed lower levels of *Mtb*-specific IgG and lower binding of *Mtb*-specific FcγR2B and FcγR3A compared to 37 TST/IGRA positive individuals[8].

In our large study in heavily exposed contacts, *Mtb*-specific antibody features (both abundance and functionality) were not different when we compared 100 IGRA-positive and 433 IGRA-negative household contacts at the time of diagnosis of the index patient. Also, no differences in baseline antibody features were seen between 51 IGRA converters and 237 persistently IGRA-negative individuals. When interpreting the antibody responses observed in our study, it is important to consider the relationship between BCG and *Mtb* antigens. While BCG shares approximately 99% genetic homology with *Mtb*[13–15], there are notable differences in antigen expression between the two organisms that influence the specificity of antibody responses. The RD1 deletion in BCG means it lacks certain antigens present in *Mtb*, such as ESAT-6 and CFP-10, making antibodies against these proteins *Mtb*-specific. However, many other antigens examined in our study are

present in both organisms. For instance, the Ag85 complex protein is produced by BCG[32,33]. Similarly, HspX (α-crystallin) is expressed by *Mtb* and, to a lesser extent, also by BCG, primarily during oxygen depletion[34,35]. Meanwhile, LAM is a cell wall component present in both mycobacterial species[36,37]. The presence of Mpt64 varies among BCG strains due to differential RD2 deletion[13]. These antigenic similarities and differences provide context for our findings that baseline antibodies recognizing *Mtb* antigens were not associated with protection against infection. The difference between our data and previous studies from the literature investigating the impact of antibodies could be due to several causes. Differences in the phenotypes of the participants ('early clearance' versus 'resisters'), our use of stricter IGRA criteria to avoid possible misclassification, or our adjustment of antibody concentrations to control measurements, may provide some explanation. Of note, the presence of a BCG-scar was associated with protection against IGRA-conversion, and BCG vaccination status interacted with innate immune correlates in household contacts, but no such relation was found between BCG vaccination and antibody profiles. Finally, intradermal BCG vaccination of adults in a low-incidence setting, which has been shown to induce trained immunity and associated with an enhanced capacity to control mycobacterial growth[38,39], only marginally changed levels of *Mtb*-specific antibodies. This is in line with older studies on BCG vaccination from Sweden, which showed protection against tuberculosis, but no significant increase in *Mtb*-specific antibodies[40].

Our study has several limitations. Our definition of *Mtb* infection was based on IGRA, which cannot distinguish mere immunoreactivity from actual infection, and which cannot identify individuals who clear *Mtb* after developing specific T-cell memory (delayed clearance). 'Early clearance', 'delayed clearance' and 'TB resisters' remain theoretical concepts affected by suboptimal tests, variable *Mtb* exposure and other factors[24,25,41,42]. However, our primary comparison was between contacts who remain IGRA-negative after 3 months, and those who convert to a positive IGRA, likely reflecting new *Mtb* infection from their recent exposure. IGRA measurements, especially with results around the standard cut-off, also show variation which could lead to misclassification, but this is unlikely with our stricter cut-offs for a negative and positive IGRA. Finally, future studies could investigate the kinetics of the immune responses over a longer period of time.

Our study also has clear strengths that allow studying correlates of protection against *Mtb* infection. We used a large cohort specifically recruited to study early clearance with follow-up of baseline IGRA-negative household contacts, we had precise estimates of *Mtb* exposure that were strongly associated with IGRA conversion and protection from BCG, and we examined both innate immune correlates and antibody features. Our findings on associations with BCG were reproduced in an independent study on BCG vaccination in a low-incidence setting. Other strengths include our optimization of signal-to-noise ratio in antibody measurements through proper filtering of antibody measurements and standardization against the positive control hemagglutinin and correction for multiple testing in all analyses.

In conclusion, our findings suggest that a more efficient host innate immune response, rather than a humoral response, mediates early clearance of *Mtb*. The protective effect of BCG vaccination against *Mtb* infection may be linked to induction of a trained immunity phenotype. Future studies should examine if induction of trained immunity can help prevention of tuberculosis in highly-exposed individuals, including in the evaluation of new TB vaccines that may offer improved protection over BCG.

## Methods
### Study design and participants
This study was embedded within a large household contact study (INFECT) which was conducted in Bandung, Indonesia, between 2014 and 2018[1]. In short, household contacts of sputum smear-positive TB patients were eligible if they were older than 5 years and had had no previous TB. They were enrolled within one week after diagnosis of the index case and screened for active TB using a symptoms screen, chest X-ray, sputum microscopy, and culture. Sociodemographic data and risk factors for *Mtb* infection were collected, including the level of exposure[1], as measured by sleeping proximity, time spent with the index patient, and presence of cavities, and sputum mycobacterial load in the index patient. Given the low prevalence of HIV among index patients (0.5%) and the general population at the time of the study (0.2%), contacts were not tested for HIV. *Mtb* infection status of contacts was assessed by QuantiFERON-TB Gold In-Tube (QFT-GIT) IGRA, which was repeated at 14 weeks in those who were initially IGRA-negative. Based on IGRA results, contacts were first classified as persistently IGRA-negative or IGRA converters, based on the IGRA test repeated after three months. To strengthen the phenotypes, instead of the manufacturer's cut-off (0.35 IU/mL), we applied stricter definitions, only including individuals whose baseline IFN-γ result (TBAg − nil tube) was <0.15 IU/mL, and whose follow-up IGRA (TBAg − nil) was either <0.15 IU/mL (persistently IGRA-negative individuals) or > 0.7 IU/mL (IGRA converters). The INFECT study was approved by the Health Research Ethics Committee of Universitas Padjadjaran Indonesia (14/UN6.C2.1.2/KEPK/PN/2014) and the Southern Health and Disability Ethics Committee New Zealand (13/STH/132).

The BCG vaccination cohort (300BCG) recruited volunteers of Western European ancestry between April 2017 and June 2018 at the Radboud University Medical Center[12,39,43-48]. Following the acquisition of written informed consent, participants underwent blood collection and then received a standard 0.1 mL dose of BCG (BCG-Bulgaria, InterVax) administered intradermally in the left upper arm by a medical doctor. The vaccination process for the study participants was conducted in groups ranging from 6 to 16 individuals each day. Blood samples were obtained two weeks and three months post-vaccination with BCG. Participants were excluded if they had been using systemic medications (excluding oral contraceptives or acetaminophen), antibiotics within three months prior to the study, a previous BCG vaccination, a history of tuberculosis, any feverish illness in the four weeks preceding the study, any vaccinations in the three months before the study, or had a medical history indicating immunodeficiency. The 300BCG (NL58553.091.16, https://onderzoekmetmensen.nl/en/trial/45603) study was approved by the Arnhem-Nijmegen Medical Ethical Committee and registered in The Overview of Medical Research in the Netherlands (OMON), formerly the Dutch Trial Register managed by Central Committee on Research Involving Human Subjects (CCMO).

### Innate immune cell phenotyping and cytokine production
Innate immune cell phenotyping with gating strategy and whole blood cytokine assays from the INFECT cohort were performed as previously described[2]. In short, we mixed heparinized blood with 123Count eBeads, followed by staining with one of three antibody panels designed to identify monocytes and granulocytes using panel 1 (CD14 AlexaFluor 488, CD16 PE, HLADR PerCP, CXCR4 APC;), innate αβ T-cells and, natural killer (NK) cells using panel 2 (CD3 AlexaFluor 488, Vα7.2 PE, CD56 PerCP, CD161 APC), and lastly, NK T cells and γδ T-cells subsets using panel 3 (CD3 AlexaFluor 488, Vα24-Jα18 PE, Vδ2 PerCP, γδ TCR APC). All antibodies from Biolegend. Details in gating strategies have been described previously[2]. Data were collected using a FACS-Calibur flow cytometer and analyzed using FlowJo software. For whole blood cytokines, samples were incubated with BCG (Danish strain 1331) $1 \times 10^5$ CFU/mL (Statens Serum Institut), *Mtb* lysate 5 µg/mL, *Streptococcus pneumoniae* (ATCC 49619) $1 \times 10^6$ CFU/mL, *Escherichia coli* $1 \times 10^6$ CFU/mL, or culture medium for 24 h at 37 °C. Supernatants were stored at − 80 °C until batchwise enzyme-linked immunosorbent assay (ELISA) measurement of tumor necrosis factor (TNF), interleukin (IL) 1β, IL-1Ra, and IL-10 (R&D Systems), IL-6, and IL-8 (Sanquin).

In the 300BCG cohort, PBMC ex vivo stimulation assays were performed as previously described[12]. PBMCs were isolated from EDTA whole blood with Ficoll-Paque (GE Healthcare) density gradient separation. PBMCs ($5 \times 10^5$) were cultured in a final volume of 200 μL/well in round-bottom 96-well plates (Greiner) and stimulated with RPMI 1640 (medium control), heat-killed *M. tuberculosis* H37Rv (5 μg/mL, specific stimulus), or heat-killed *S. aureus* ($1 \times 10^6$ CFU/mL, non-specific stimulus). Supernatants were collected after 24 h and 7 days of incubation at 37 °C and stored at – 20 °C until analysis. Cytokine levels were measured at 24 h (IL-1β, IL-6, and TNF) and 7 days (IFN-γ). Supernatant samples from all time points for a participant were measured on the same plate to ensure that variation between plates would not affect the calculated fold changes.

## IGRA supernatant inflammatory marker measurements

Inflammatory proteins from IGRA supernatant nil and mitogen tube (PHA stimulation) were measured using the commercially available Olink Proteomics AB Inflammation Panel (92 inflammatory proteins) (Uppsala Sweden). In this assay, proteins are recognized by antibody pairs coupled to cDNA strands which bind in close proximity, followed by extension by a polymerase reaction. Quality control was performed by Olink Proteomics, with 8% of samples not passing the quality control and subsequently excluded from the analysis. We only analyzed proteins detected in 75% of individuals. Overall, 67 of the 92 (81.5%) proteins were detected in at least 75% of the plasma samples and included in the analysis.

## Antibody measurements

For antibody assays, *Mtb* antigens tested were: purified protein derivative (PPD) (Statens Serum Institute), Psts-1 (BEI Resources Cat #NR-53528), Tbad (in-house prepared, see PMID: 31427817), Apa (BEI Resources Cat # NR-14862), Mpt 64 (BEI Resources Cat # NR-49435), Ag85A and B in a 1:1 ratio (BEI Resources Cat#NR-49427 and #NR-53526), recombinant ESAT-6 (BEI Resources Cat#NR-49424) and CFP-10 (BEI Resources Cat#NR-49425) in a 1:1 ratio, HspX (BEI Resources Cat#NR-49428), and lipoarabinomannan (LAM) (BEI Resources Cat#NR-14848). An equal mixture of influenza antigens from HA1(B/Brisbane/60/2008) and HA1 (A/New Caledonia/20/99) (Immune Technology Corp ITIT-003-001p and IT-003-B3p) was used as a positive assay control. A Luminex assay was used to quantify the relative levels of antigen-specific antibody isotypes and subclasses and their ability to bind Fc receptors. Luminex Magplex carboxylated microspheres (Luminex Corporation) were coupled to proteins/antigens via covalent N-hydroxysuccinimide (NHS)−ester linkages by 1-ethyl-3-(3-dimethylaminopropyl)carbodiimide hydro-chloride (EDC) and sulfo-NHS per manufacturer recommendations. LAM was modified by 4-(4,6-dimethoxy [1,3,5]triazin-2-yl)-4-methyl-morpholinium (DMTMM) prior to conjugation. Individual microsphere with unique fluorescence regions allowed for multiplexed flow cytometry-based quantifications[49].

Diluted serum samples were incubated with pooled microspheres for 16 h at room temperature and then washed three times with 0.1% bovine serum albumin (BSA)/0.05% Tween-20 in PBS. Secondary incubations were performed for 2 h at room temperature. Then, samples were washed three times prior to acquisition. For each assay, the median fluorescence intensity (MFI) for each bead region was measured using an iQue Plus Screener (Intellicyt). For the detection of FcγR-binding antibodies, diluted serum samples were incubated with the antigen-coated beads as above. For detection, PE-labeled Strepavidin was coupled to biotinylated, purified FcγRs (Duke Human Vaccine Institute). Excess D-desthiobiotin was used to saturate unbound Strep-PE. The Strep-FcγR was then diluted in 0.1 % BSA, 0.05 % Tween-20, and 1X PBS. The blocked detection reagent was then added as a secondary step similar to above, and MFI for each bead region was quantified using an iQue Plus Screener (Intellicyt).

## Data analysis and statistics

All computational analyses were performed in R 4.2.3 with an Rstudio integrated development environment[50,51]. Figures were generated using the R package 'ggplot2' or 'ggpubr' unless stated otherwise[52]. Tables were made using the R package 'gtsummary'[53]. For all comparisons, we used two-sided statistical testing.

To compare cell subpopulations in the INFECT cohort across different time periods, $\log_{10}$-transformed cell counts at 2 weeks and 14 weeks were calculated for each participant. Unpaired Mann-Whitney U tests were used to compare cell subpopulations between groups at week 2[52]. Paired Wilcoxon signed rank tests was used for paired significance comparisons of transformed cell counts in week 2 and week 14[52]. Both calculations were adjusted using Benjamini-Hochberg (false discovery rate). The median fold change and 95% confidence interval calculated on untransformed cell counts are also presented. The fold change between IGRA converters and persistently IGRA-negatives was compared using an unpaired Mann-Whitney U test. The same was done to show the median fold change and 95% confidence interval between persistently IGRA-negative with BCG scar and without BCG scar. A decrease in cell count is indicated by a fold change of less than 1. The median fold change and confidence interval were calculated using the MedianCI function from 'DescTools' R package[54]. Paired Wilcoxon signed-rank tests were used, and the *P*-value was adjusted for multiple testing using Benjamini-Hochberg.

For cytokine measurements in the INFECT cohort, concentrations below the detection limit were substituted with the lowest detectable limit for each cytokine (39 pg/mL for TNF, 19.5 pg/mL for IL-1β, 195 pg/mL for IL-1Ra, 312 pg/mL for both IL-6 and IL-8 and 4.68 pg/mL for IL-10); the highest number for which this was done was for *Mtb* induced TNF production (3%). Contaminated samples, defined as samples with detectable IL-6 in unstimulated samples, were removed from the analysis. Cytokine data were $\log_{10}$ transformed. Batch effects were removed using the RemoveBatchEffect function from 'limma'[55], and analyses were carried out on the residuals from this model fit. Heatmaps were created using the 'ComplexHeatmap' package[56] visualizing the median Z-score of the batch-adjusted cytokine variables. Unpaired Mann-Whitney *U* tests were used to compare adjusted cytokine levels between groups. Logistic regression was used to estimate the associations between cytokine production and IGRA status, using $\log_{10}$ transformed cytokine measurements and adjusting for age, sex, BMI, blood monocyte count, blood lymphocyte count, and batch. The association of cytokines with IGRA status at follow-up was also adjusted for exposure; this could not be done at baseline since exposure risk scores were unavailable for contacts with a positive IGRA at baseline. Odds ratios were calculated from the beta estimates and adjusted for multiple tests using Benjamini-Hochberg.

For inflammatory proteins, only samples and proteins that passed quality control were used for the analysis. As protein measurements, especially in low concentration, can be affected by hemolysis, we excluded proteins that might be impacted by hemolysis of less than 3.8 g/L based on the Olink Inflammatory Protein validation data sheet. We also excluded samples that had hemolysis of more than 15 g/L (as determined by two researchers blinded to IGRA status independently visually matching the sample to the hemolysis concentration reference in the Olink validation data sheet). The inflammatory protein relative levels (NPX) were $\log_2$ transformed. Logistic regression models were used to estimate the association between NPX measurement of each inflammatory protein at baseline and IGRA status at follow-up adjusting for age, sex, BMI, and exposure risk score. In addition, linear regression was used to find the correlation between inflammatory protein levels with quantitative IGRA IFN-γ (TBAg − Nil) levels at follow-up.

For analysis of antibody profiles, for each individual, we divided anti-*Mtb* antibody levels by the level of hemagglutinin (HA)-specific

antibody to adjust for non-specific interindividual variation in antibody production and increase the specificity of anti-*Mtb* antibodies. These ratios were $\log_{10}$ transformed. We established a lower limit of quantification for each antigen as the mean MFI + 6 SD (standard deviation) in the PBS control. For statistical comparisons of antibody profiles by IGRA status, we used unpaired Mann-Whitney U tests, corrected for multiple testing by a Benjamini-Hochberg, and showed the fold change in the heatmap. Supervised clustering using partial least squares discriminant analysis (PLS-DA) using the 'mixOmics' package on Z-scored data was used to discriminate the antibody profile explained by IGRA status, both at baselines and at follow-up[57]. Logistic regression models adjusting for age, sex, and BMI were used to find the associations between antibody levels and IGRA status. Exposure risk score was added for regression analysis linking baseline antibody levels and IGRA results at follow-up, but not at baseline since exposure risk scores were not available for baseline IGRA-positive individuals. Functional antibody variables (antibody-dependent complement deposition, antibody-dependent cellular phagocytosis, and antibody-dependent neutrophil phagocytosis) specific to LAM were compared using the unpaired Mann-Whitney *U* tests. In addition, logistic regression adjusting for age, sex, BMI, and exposure risk score was used to estimate associations between antibody functionality and IGRA status at follow-up.

In the 300BCG cohort, ex vivo cytokine measurements were $\log_{10}$ transformed and corrected for batch effect using linear regression[45]. The heatmap of fold change between pre-vaccination and day 90 post-vaccination was shown. Paired Wilcoxon signed-rank tests were used for statistical comparisons of the pre-vaccination and 90-day post-vaccination ex vivo cytokine levels. Antibody MFI were standardized to the MFI of HA-specific antibodies as above. The ratios were then $\log_{10}$ transformed. The heatmap of the fold change of antibody level between pre-vaccination and 90 days post-vaccination was shown. Paired Wilcoxon signed-rank tests were used for statistical comparisons of the pre-vaccination and 90-day post-vaccination.

### Reporting summary
Further information on research design is available in the Nature Portfolio Reporting Summary linked to this article.

## Data availability
The data supporting the findings of this study are available in the accompanying Supplementary Information. Source data are provided in this paper.

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

## Acknowledgements

The authors extend their appreciation to the dedicated teams involved in fieldwork, laboratory activities, and data management, including the recruitment of the INFECT cohort. This team comprised Andini Cahya Nurani, Novianti, Deni, Wiwik Pratiwi, Dody Taufik Akbar, Emira Diandini, Dwi Febni Ratnaningsih, Inas Kathina, Yusak Sastra Atmaja, Nuni Haeruni, Anbarunik Puteri Danthin, Harold Eka Atmaja, Alif Al Birru, Nopi Susilawati, and Runi Rahmawati. Also, Rachel F. Hannaway, who works on INFECT from Otago University. Special thanks are also due to the TANDEM study team, especially coordinator Raspati C. Koesoemadinata, Lidya Chaidir, Jessi Annisa, and Ria Windyani for their cooperation. In addition, the authors acknowledge Corina van den Heuvel, Heidi Lemmers, and Helga Dijkstra for their assistance with the ELISA procedures. Also, we would like to extend our thanks to Liesbeth van Emst for her help in Olink measurements. We would also like to thank all volunteers from the 300BCG cohort for participation in the study. A.V.J. was supported by a New Zealand Health Research Council Clinical Training Research Fellowship. INFECT cohort recruitment was funded by the University of Otago and Mercy Hospital (through an endowment fund and directly), Dunedin, New Zealand. Index case recruitment and investigation was part of the TANDEM project (www.tandem-fp7.eu), which is supported by the European Union's Seventh Framework Program (FP7/2007–2013) under grant agreement number 305279. The IGRA (QuantiFERON) was donated by Qiagen. Flow cytometry analysis was supported by a grant from the Dean's Bequest Fund, University of Otago. R.P.M. and the Systems Serology Laboratory are supported by the generous gifts of Terry and Susan Ragon, and Mark and Lisa Schwartz. R.P.M. receives funding from the Global Health Vaccine Accelerator Program (GH-VAP) through the Bill and Melinda Gates Foundation (INV-001650). R.v.C. was supported by the Royal Netherlands Academy of Arts and Sciences (09-PD-14) and the VIDI grant 017.106.310 of The Netherlands Organization for Scientific Research. M.G.N. was supported by an ERC advanced grant (833247) and a Spinoza grant from The Netherlands Organization for Scientific Research. L.C.J.D.B. was partly funded by a grant to the Research Center for Vitamins and Vaccines (CVIVA) from the Danish National Research Foundation (DNRF108).

## Author contributions

Conceptualization: T.P.S., G.A., V.A.C.M.K., and R.v.C.; Methodology: T.P.S., V.A.C.M.K., and R.v.C.; Data curation: T.P.S.; Formal analysis: T.P.S. and P.P.H.; Investigation: T.P.S., L.A., A.J.V., F.U., M.S., J.E.U., K.S., P.K., H.M., J.S.L., V.A.C.M.K., S.J.C.F.M.M., L.C.J.D.B., V.P.M., and L.A.B.J.;

Resources: P.C.H., B.A., and R.v.C.; Writing – original draft: T.P.S. and R.v.C.; Writing – review & editing: K.S., J.U., R.P.M., A.v.L., A.R.I, L.A.B.J., P.C.H., M.G.N., V.A.C.M.K., and R.v.C.; Visualization: T.P.S. and P.P.H.; Supervision: M.G.N., V.A.C.M.K., and R.v.C.

## Competing interests

The authors declare no competing interests.

## Inclusion & ethics

All collaborators of this study have fulfilled the criteria for authorship required by Nature Portfolio journals and have been included as authors. The research included Indonesian researchers throughout the research process, as their participation was essential for the design and implementation of the study. Capacity-building plans for Indonesian researchers were discussed and have been implemented, with T.P.S. and L.A. exemplifying the success of these efforts within our long-standing collaboration. Roles and responsibilities were agreed among collaborators ahead of the research. This work includes findings that are locally relevant which have been determined in collaboration with local partners. This research was not severely restricted or prohibited in the setting of the researchers and does not result in stigmatization, incrimination, discrimination or personal risk to participants. The study has been approved by a local ethics review committee. Local and regional research relevant to our study was taken into account in citations. This research does not involve health, safety, security or other risk to researchers. Benefit-sharing measures have been discussed among all collaborators regarding biological materials transferred out of Indonesia.

## Additional information

[1]Department of Internal Medicine and Radboud Community for Infectious Diseases (RCI), Radboud University Medical Center, Nijmegen, the Netherlands. [2]Research Center for Care and Control of Infectious Diseases, Universitas Padjadjaran, Bandung, Indonesia. [3]Department of Public Health, Faculty of Medicine, Universitas Padjadjaran, Bandung, Indonesia. [4]Department of Pathology and Molecular Medicine, University of Otago, Wellington, New Zealand. [5]Department of Microbiology and Immunology, University of Otago, Dunedin, New Zealand. [6]Department of Clinical Pathology, Faculty of Medicine, Universitas Padjadjaran, Bandung, Indonesia. [7]Faculty of Medicine, Universitas Indonesia, Jakarta, Indonesia. [8]Ragon Institute of Mass General, MIT, and Harvard, Cambridge, Massachusetts, USA. [9]Department of Medical Genetics, Iuliu Hatieganu University of Medicine and Pharmacy, Cluj-Napoca, Romania. [10]Department of Immunology and Metabolism, Life and Medical Sciences Institute, University of Bonn, Bonn, Germany. [11]Department of Internal Medicine, Faculty of Medicine, Universitas Padjadjaran, Bandung, Indonesia. [12]Department of Immunology and Infectious Diseases, Harvard T.H. Chan School of Public Health, Boston, Massachusetts, USA. [13]Department of Mathematics and Statistics, University of Otago, Dunedin, New Zealand. [14]Research Centre Innovations in Care, Rotterdam University of Applied Sciences, Rotterdam, the Netherlands. [15]Centre for International Health, University of Otago, Dunedin, New Zealand. [16]Centre for Tropical Medicine and Global Health, Nuffield Department of Medicine, University of Oxford, Oxford, UK. ✉e-mail: Reinout.vanCrevel@radboudumc.nl

