## [Peer Review file · Nature Communications]

Immune correlates of early clearance of *Mycobacterium tuberculosis* among tuberculosis household contacts in Indonesia

Corresponding Author: Professor Reinout van Crevel

Version 0:

Reviewer comments:

Reviewer #1

(Remarks to the Author)

The authors have comprehensively addressed the reviewers comments. The revised manuscript is much improved and clearer. It also stands better alone without reference to the previous published work in JID than it did. The purpose of the second cohort is also much clearer now.

Reviewer #2

(Remarks to the Author)

This manuscript reports investigation of immune response differences between individuals who live in households with persons with TB and are stratified by 3m longitudinal differences in acquisition of T cell memory to Mtb antigens contained within the QFT test. The underlying premise is that those who do not develop T cell memory to the antigens in the QFT test within the 3 months of testing demonstrate a phenotype of early clearance of Mtb upon infection. In contrast, those who develop T cell memory are assumed to not clear the infection as quickly. This is because those who develop T cell memory are more likely to develop TB, thus it is inferred they are more likely to remain infected, however only a minority (<5%) are likely to develop TB within 12m, thus how many of those who are QFT positive at 3m would have also cleared the infection is unknown. It is therefore an assumption of early clearance of infection, but how much earlier and whether there is a true difference in speed of clearance given the same level of exposure, is impossible to measure. It could be a same level of clearance via different mechanisms (given antibodies are still measurable in IGRA negative, so could be inferred to indicate they were equally infected at least for sufficient time to develop antibody responses). Given it is impossible to measure whether earlier clearance actually occurred, caution should be taken not to over infer interpretation of the QFT results and instead just report based on the result measured.

The authors have revised the manuscript based on initial reviewers' comments. This has improved the manuscript and the revisions have addressed some of the major concerns, including adjusting for exposure risk score wherever possible, given that those who remained negative had lower exposure scores. The major concerns that remain are:

1. The main finding is that those with a BCG scar from BCG vaccination at birth are more likely to remain QFT negative. Immune cell responses and humoral differences in the recent TB contacts are compared between those with and without BCG scar. The conclusion is that BCG at birth impacts difference in innate immune cell responses and not humoral responses, to recent Mtb infection. The direct effect of BCG vaccination is then investigated in the absence of recent Mtb exposure in a second cohort, and humoral responses to Mtb measured, as well as innate responses to Mtb and heterologous antigens.

The issue is that the first cohort is investigating the distant effect of BCG vaccination at birth on subsequent responses to Mtb infection. The second cohort only measures the response to BCG, immediately after vaccination. It still needs to be better explained how this second cohort is validating observations from the first, with direct relevance to the primary objective of

understanding immune responses associated with Mtb early clearance following recent infection.

2. As the second cohort had no prior Mtb infection before receiving BCG it is unclear why the effect of BCG on boosting Mtb-specific antibodies was measured, when these antibodies shouldn't be detected. In response to the previous reviewer comments, the authors suggested "Some unspecific binding in the antibody assays seems a more likely explanation for the positive results." If these Mtb-specific antibodies results are non-specific then they should not be reported as a major finding in the abstract and I would suggest removing these from the manuscript. As IGRA negativity in the first cohort was not associated with humoral responses, there isn't strong support to measure it in the second cohort. The only thing that could be reasonably measured is the effect of BCG on heterologous pathogen humoral responses to which the individuals would likely be exposed. Were the same antigens and antibody assays used for each cohort?

3. Longitudinal significant differences were also reported for these Mtb-specific antibodies in the second cohort. Given that these responses are likely non-specific, this longitudinal significant change may be a consequence of the normalisation approach, due to seasonal differences in anti-HA resulting in a significant difference in the corrected ratio. Plotting the raw anti-HA values, instead of the anti-HA/anti-HA ratio would provide some insight. If there are longitudinal differences in raw anti-HA MFI then this isn't a good control for normalisation in the second cohort as there isn't a control group for comparison to those who received BCG.

4. As mentioned in the previous review, the authors should take care not to conflate the absence of a BCG scar with not receiving BCG vaccination. Therefore when reporting results for the first cohort, 'BCG scar' should always be used (ie line 96-97) and not replaced with BCG vaccination. When BCG Scar is first mentioned on line 64, it would be beneficial to indicate here that this occurs due to BCG vaccination at birth, but not all vaccinated people will develop a scar.

5. The description of the cohort at the start of the results should include age ranges and numbers of children and adults included. This is important given the relationship between age, BCG scar and IGRA conversion. Children received BCG vaccination much more recently than adults.

6. In response to reviewers authors state that baseline cytokine and IGRA relationship results cannot be adjusted for exposure score as this data was not collected for those IGRA positive at baseline, but lines 170-171 indicates it was collected? Please clarify. Table S2 should refer to Table S3, I think.

7. Flow cytometry methods need additional detail on the antibody panels used, controls, analysis and gating strategy.

Minor suggestions

- Line 65, should be BCG'-associated' protection
- Line 72, add 'recently' to heavily exposed
- Line 112, the genotype -dependent difference in protection conferred by BCG vaccination [change to SCAR] , should include description that this is related to the 'other lineages', given that the prior sentences relate to Beijing lineages and the current wording may imply the relationship is with Beijing. Given the WHO guideline to move away for geographical naming of pathogens to avoid stigmatisation, it is suggested to use lineage number terminology instead of Beijing here.
- Line 133, New results refer to frequencies of cell populations after exposure score adjustment. The preceding results in that paragraph all describe results relating to cell number, not frequencies. Are cell numbers similarly not affected by exposure?
- Line 171, should refer to antibody ratio, not concentrations.
- All figures should ensure Y-axis is labelled as MFI/HA MFI if the ratio is plotted, not MFI.
- Line 387, was Mtb live or dead?
- Line 417, additional antigens need to be included, not all antigens are listed for antibodies measured.
- Table S1, subset columns would be better labelled as 'strict cut-off'

Version 1:

Reviewer comments:

Reviewer #2

(Remarks to the Author)

The authors have responded to all of my comments and revised the manuscript accordingly. Our difference in opinion remains on the two points below. The first point I am happy to accept as a difference in opinion, given they have included text to address my main concern regarding the nuances of IGRA interpretation. The second I have made a suggestion for consideration, but will leave it to the authors to interpret these finding, with my concerns noted.

1. Early clearance terminology - I appreciate the authors adding extra text to emphasise the theoretical conceptual interpretation of IGRA results, given that true early clearance cannot be determined. I remain more hesitant than the authors in over-interpreting the results of a negative 3m IGRA as indicating early clearance, particularly without testing for alternate antigen T cell responses, given a greater breadth of humoral antigen responses were measured (and detected). I accept this as a difference in opinion that I have with others in the field, resulting from the limitation in the tools we have as a field.

2. Mtb-specific antibodies following recent BCG - Thank you for including data confirming there is no longitudinal anti-HA difference. We will continue to disagree on the point regarding Mtb-specific antibodies. It is the authors prerogative to include assessment of antibody responses that are non-specific to the pathogen challenge to investigate heterologous responses,

as long as that is how they are interpreted. I think it needs to be explicitly stated that whilst BCG may be 99% homologous to Mtb, BCG contains many differences, including the RD1 deletion so it does not contain ESAT-6, CFP-10 or Ag85A/B and therefore should not induce these antibodies directly. Therefore finding significant changes in these antibodies following BCG, points to modifying non-specific humoral responses. Perhaps the authors would consider revising results to specify which antibodies are detecting antigens present in BCG and which antibodies are detecting heterologous responses?

I also suggest harmonising language regarding these results, which are currently conflicting. In the results, lines 222-224 states "several Mtb specific IgG3 showed a statistically significant, albeit minimal increase, while several Mtb-specific IgM antibodies showed a minimal decrease. The discussion states in lines 240-241 "in a low incidence setting, adult BCG vaccination induced heterologous cytokine production, but did not lead to significant changes in anti-Mtb antibodies; in lines 321-324, "intradermal BCG vaccination of adults in a low-incidence setting..... did not significantly alter titers of Mtb-specific antibodies." The abstract states, " BCG vaccination induced heterologous innate cytokine production, but only marginally affected Mtb-specific antibody profiles".

Minor comments:

L 80-82 - sentence doesn't make grammatical sense.

L 93 - does 'respectively' come after 'years of age'?

L 143 - indicate 'Mtb lysate', as you have now indicated in the methods.

L 242-245 - these abbreviations should have been in the intro already

Reviewer #1:

The authors have comprehensively addressed the reviewers comments. The revised manuscript is much improved and clearer. It also stands better alone without reference to the previous published work in JID than it did. The purpose of the second cohort is also much clearer now.

Thank you very much.

Reviewer #2:

This manuscript reports investigation of immune response differences between individuals who live in households with persons with TB and are stratified by 3m longitudinal differences in acquisition of T cell memory to *Mtb* antigens contained within the QFT test. The underlying premise is that those who do not develop T cell memory to the antigens in the QFT test within the 3 months of testing demonstrate a phenotype of early clearance of *Mtb* upon infection. In contrast, those who develop T cell memory are assumed to not clear the infection as quickly. This is because those who develop T cell memory are more likely to develop TB, thus it is inferred they are more likely to remain infected, however only a minority (<5%) are likely to develop TB within 12m, thus how many of those who are QFT positive at 3m would have also cleared the infection is unknown. It is therefore an assumption of early clearance of infection, but how much earlier and whether there is a true difference in speed of clearance given the same level of exposure, is impossible to measure.

*The term 'early clearance' has been used by others, such as a 2016 Lancet review on TB transmission¹, but both early and delayed clearance are theoretical concepts, that are affected by suboptimal tests, variable *Mtb* exposure and other factors. And indeed, early clearance cannot be easily distinguished from delayed clearance (outside the scope of our study).*

*We have now added “ and which cannot identify individuals who clear *Mtb* after developing specific T-cell memory (delayed clearance). 'Early clearance',*

‘delayed clearance’ and ‘TB resisters’ remain theoretical concepts affected by suboptimal tests, variable *Mtb* exposure and other factors¹⁻⁴.” (line 324-328)

It could be a same level of clearance via different mechanisms (given antibodies are still measurable in IGRA negative, so could be inferred to indicate they were equally infected at least for sufficient time to develop antibody responses).

The fact that there were no significant differences in antibody concentrations between baseline IGRA-positive and –negative individuals points against an important role of antibodies in preventing infection (lines 178-183) .

Given it is impossible to measure whether earlier clearance actually occurred, caution should be taken not to over infer interpretation of the QFT results and instead just report based on the result measured.

We distinguish ‘persistently IGRA negative status’ from ‘IGRA conversion’ (with stricter criteria to avoid misclassification) and based on two IGRAs with a 3-month interval to study a particular exposure in one household, and separating it from the ‘TB resister’ phenotype, and used quantitative IGRA results where possible.

Other reviewers appreciated the strength of our phenotypes and analysis. IGRA conversion at 3 months strongly correlated with sputum bacterial load and other measures of exposure; strain virulence (Beijing strains); and BCG scar status, and we were able to adjust analysis of immunological characteristics for these factors (and age and sex). We believe it is reasonable to keep the label of ‘early clearance’ (also used in reviews written by others¹) but we have added further nuance in the discussion (line 326-330)

The authors have revised the manuscript based on initial reviewers' comments. This has improved the manuscript and the revisions have addressed some of the major concerns, including adjusting for exposure risk score wherever possible, given that those who remained negative had lower exposure scores.

Thank you

The major concerns that remain are:

1. The main finding is that those with a BCG scar from BCG vaccination at birth are more likely to remain QFT negative. Immune cell responses and humoral differences in the recent TB contacts are compared between those with and without BCG scar. The conclusion is that BCG at birth impacts difference in innate immune cell responses and not humoral responses, to recent *Mtb* infection. The direct effect of BCG vaccination is then investigated in the absence of recent *Mtb* exposure in a second cohort, and humoral responses to *Mtb* measured, as well as innate responses to *Mtb* and heterologous antigens. The issue is that the first cohort is investigating the distant effect of BCG vaccination at birth on subsequent responses to *Mtb* infection. The second cohort only measures the response to BCG, immediately after

vaccination. It still needs to be better explained how this second cohort is validating observations from the first, with direct relevance to the primary objective of understanding immune responses associated with *Mtb* early clearance following recent infection.

*The reviewer rightly points to an important difference between the two cohorts. The Indonesian cohort had received BCG-vaccinated at birth and was examined years afterwards, the second cohort of adults in a low-endemic setting was studied before and 3 months after a BCG vaccination (their first). In the Indonesian cohort, IGRA conversion was much less common among subjects with a BCG scar (RR 0.35; 95% CI 0.21-0.58), and BCG scars were associated with more pronounced differences in immune cell phenotype and function between persistently IGRA-negative individuals and IGRA converters (Fig 1C and 2F). Because of these findings in Indonesia, we employed the second cohort in a low-burden setting to investigate the effects of BCG vaccination on immune responses in adults without prior exposure. We confirmed effects of BCG on heterologous cytokine production, and the absence of an effect on anti-*Mtb* antibodies.*

We've further clarified this in line 206-213.

2. As the second cohort had no prior *Mtb* infection before receiving BCG it is unclear why the effect of BCG on boosting *Mtb*-specific antibodies was measured, when these antibodies shouldn't be detected.

*We examined the effect of vaccination with BCG (an attenuated form of *M. bovis*, which shares 99% genetic similarity with *Mtb*)^{5,6} on induction (rather than boosting) of antibodies. We now clarify in line 206-215:*

*“To further investigate the induction of innate immune responses and antibody production after mycobacterial stimulation in vivo, we next used a cohort of healthy volunteers vaccinated with BCG in a low-TB incidence setting¹². Presence of a BCG scar had shown strong relations with immune markers among individuals in a high-burden setting (Indonesia) who were examined years after they had been vaccinated at birth, and we went to a low-burden setting to examine this effect of BCG (before and three months after vaccination) in the absence of possible confounding by exposure to *M. tuberculosis*.*

*As expected, vaccination with BCG (an attenuated form of *M. bovis* which shares 99% similarity with *Mtb*)“*

And in line 219:

*“To examine if BCG vaccination induced anti-*Mtb* antibodies,”*

In response to the previous reviewer comments, the authors suggested "Some unspecific binding in the antibody assays seems a more likely explanation for the positive results." If these *Mtb*-specific antibodies results are non-specific then they should not be reported as a major finding in the abstract and I would suggest removing these from the manuscript.

This problem of incomplete specificity of antibodies is not unique for anti-Mtb antibodies. By adjusting for anti-HA antibody titers (present in 100% of individuals) we have tried to increase specificity. We have now underlined this in line 517-518.

*Since there has been so much interest in antibodies and resistance to *Mtb* infection, we would prefer to keep the antibody data from our study (the largest cohort reported so far, and the only one looking at early time points after household exposure and experimental BCG vaccination in adults) in our manuscript.*

As IGRA negativity in the first cohort was not associated with humoral responses, there isn't strong support to measure it in the second cohort. The only thing that could be reasonably measured is the effect of BCG on heterologous pathogen humoral responses to which the individuals would likely be exposed. Were the same antigens and antibody assays used for each cohort?

We may not have been clear enough in our justification of the 2nd cohort. Individuals in the first cohort were enrolled many years after BCG vaccination. We used the second cohort to look at induction of antibodies under experimental conditions, 3 months after BCG vaccination. This is much stronger evidence that BCG vaccination does not impact antibody concentrations, while it does affect innate immune characteristics. We've now further clarified the justification for looking at antibodies in the second cohort. See line 208-212 and line 218.

3. Longitudinal significant differences were also reported for these *Mtb*-specific antibodies in the second cohort. Given that these responses are likely non-specific, this longitudinal significant change may be a consequence of the normalisation approach, due to seasonal differences in anti-HA resulting in a significant difference in the corrected ratio. Plotting the raw anti-HA values, instead of the anti-HA/anti-HA ratio would provide some insight. If there are longitudinal differences in raw anti-HA MFI then this isn't a good control for normalisation in the second cohort as there isn't a control group for comparison to those who received BCG.

Below we have plotted raw anti-HA IgM and IgG3 MFI. There are no longitudinal differences in anti-HA IgM and IgG3 MFI before and 90 post-vaccination. Yet, we found significant in anti-Mtb IgG3 and IgM antibodies after normalization to anti-HA. Thus, anti-HA is a good normalization and we are confident that the differences in the anti-Mtb antibodies responses are indeed specific. We have added these plots below in Fig. 5D. See also line 224.

4. As mentioned in the previous review, the authors should take care not to conflate the absence of a BCG scar with not receiving BCG vaccination. Therefore when reporting results for the first cohort, 'BCG scar' should always be used (ie line 96-97) and not replaced with BCG vaccination. When BCG Scar is first mentioned on line 64, it would be beneficial to indicate here that this occurs due to BCG vaccination at birth, but not all vaccinated people will develop a scar.

We agree, thank you. We've now adjusted this throughout the manuscript and in the abstract and have added: "It should be noted that not all individuals develop a scar after BCG vaccination⁷." in line 67-68.

5. The description of the cohort at the start of the results should include age ranges and numbers of children and adults included. This is important given the relationship between age, BCG scar and IGRA conversion. Children received BCG vaccination much more recently than adults.

We're not sure we understand what the reviewer means. BCG vaccination does not lead to a positive IGRA, as IGRA (purposely) use antigens that are not present in BCG. There is likely some relation between age and BCG scar positivity, and of course between age and baseline IGRA status (as in Table 1). We've now added information on age in the results (line 90-92).

6. In response to reviewers authors state that baseline cytokine and IGRA relationship results cannot be adjusted for exposure score as this data was not collected for those IGRA positive at baseline, but lines 170-171 indicates it was collected? Please clarify. Table S2 should refer to Table S3, I think.

What we meant with “higher exposure to the index case” in line 170-171 was that baseline IGRA-positives more often slept in the same room as the index case, spent more hours in contact with them, and had a higher likelihood of living with an index case with cavitory disease on chest X-ray. These variables are the component of exposure to the index case. We clarified this in the updated line 173-175.

Yes! it should refer to Table S3, thank you.

7. Flow cytometry methods need additional detail on the antibody panels used, controls, analysis and gating strategy.

We added additional detail in the method section (line 393-401).

Minor suggestions:

- Line 65, should be BCG'-associated' protection

See updated sentence in line 65.

- Line 72, add 'recently' to heavily exposed

We've added this. Line 73.

- Line 112, the genotype -dependent difference in protection conferred by BCG vaccination [change to SCAR] , should include description that this is related to the 'other lineages', given that the prior sentences relate to Beijing lineages and the current wording may imply the relationship is with Beijing. Given the WHO guideline to move away for geographical naming of pathogens to avoid stigmatisation, it is suggested to use lineage number terminology instead of Beijing here.

Now we opted to first introduce “L2 (Beijing) strain” to accommodate both the lineage number terminology and the more widely known geographical naming and then continue with lineage number terminology in the main text. See line 109, 111, 112, 262, and Table 1.

- Line 133, New results refer to frequencies of cell populations after exposure score adjustment. The preceding results in that paragraph all describe results relating to cell number, not frequencies. Are cell numbers similarly not affected by exposure?

Sorry, we meant cell numbers (cell number/ μ L), not frequencies (%). See updated sentence in line 136.

- Line 171, should refer to antibody ratio, not concentrations.

Agree. See line 175.

- All figures should ensure Y-axis is labelled as MFI/HA MFI if the ratio is plotted, not MFI.

Done.

- Line 387, was *Mtb* live or dead?

Dead. *Mtb* whole cell lysate. See line 404.

- Line 417, additional antigens need to be included, not all antigens are listed for antibodies measured.

Done. See updated paragraph in line 433.

- Table S1, subset columns would be better labelled as 'strict cut-off'

Agree. Done. See Table S1.

Reference

1. Yates, T. A. *et al.* The transmission of *Mycobacterium tuberculosis* in high burden settings. *The Lancet Infectious Diseases* **16**, 227–238 (2016).
2. Emery, J. C. *et al.* Self-clearance of *Mycobacterium tuberculosis* infection: implications for lifetime risk and population at-risk of tuberculosis disease. *Proceedings of the Royal Society B: Biological Sciences* **288**, 20201635 (2021).
3. Simmons, J. D. *et al.* Immunological mechanisms of human resistance to persistent *Mycobacterium tuberculosis* infection. *Nature Reviews Immunology* **18**, 575–589 (2018).
4. Verrall, A. J., Netea, M. G., Alisjahbana, B., Hill, P. C. & van Crevel, R. Early clearance of *Mycobacterium tuberculosis*: a new frontier in prevention. *Immunology* **141**, 506–513 (2014).
5. Mahairas, G. G., Sabo, P. J., Hickey, M. J., Singh, D. C. & Stover, C. K. Molecular analysis of genetic differences between *Mycobacterium bovis* BCG and virulent *M. bovis*. *Journal of Bacteriology* **178**, 1274–1282 (1996).
6. Asadian, M., Hassanzadeh, S. M., Safarchi, A. & Douraghi, M. Genomic characteristics of two most widely used BCG vaccine strains: Danish 1331 and Pasteur 1173P2. *BMC Genomics* **23**, 609 (2022).
7. Villanueva, P. *et al.* Factors influencing scar formation following Bacille Calmette-Guérin (BCG) vaccination. *Heliyon* **9**, e15241 (2023).

Reviewer #2 (Remarks to the Author):

The authors have responded to all of my comments and revised the manuscript accordingly. Our difference in opinion remains on the two points below. The first point I am happy to accept as a difference in opinion, given they have included text to address my main concern regarding the nuances of IGRA interpretation. The second I have made a suggestion for consideration, but will leave it to the authors to interpret these findings, with my concerns noted.

1. Early clearance terminology - I appreciate the authors adding extra text to emphasise the theoretical conceptual interpretation of IGRA results, given that true early clearance cannot be determined. I remain more hesitant than the authors in over-interpreting the results of a negative 3m IGRA as indicating early clearance, particularly without testing for alternate antigen T cell responses, given a greater breadth of humoral antigen responses were measured (and detected). I accept this as a difference in opinion that I have with others in the field, resulting from the limitation in the tools we have as a field.

Answer: We thank the reviewer for this nuance, and we 'agree to disagree' ↻

2. Mtb-specific antibodies following recent BCG - Thank you for including data confirming there is no longitudinal anti-HA difference. We will continue to disagree on the point regarding Mtb-specific antibodies. It is the authors prerogative to include assessment of antibody responses that are non-specific to the pathogen challenge to investigate heterologous responses, as long as that is how they are interpreted. I think it needs to be explicitly stated that whilst BCG may be 99% homologous to Mtb, BCG contains many differences, including the RD1 deletion so it does not contain ESAT-6, CFP-10 or Ag85A/B and therefore should not induce these antibodies directly. Therefore finding significant changes in these antibodies following BCG, points to modifying non-specific humoral responses. Perhaps the authors would consider revising results to specify which antibodies are detecting antigens present in BCG and which antibodies are detecting heterologous responses?

Answer: True, ESAT-6 and CFP-10 are specific for Mtb because of the loss of RD1 in BCG. But most of the other antigens are actually also produced by BCG. For example, Ag85 is also known to be produced by BCG^{1,2}. HspX (or alpha-crystallin) is expressed by BCG, albeit only in small amounts during oxygen depletion^{3,4}. LAM is present in both Mtb and BCG⁵. The presence of Mpt64 in BCG is variable in different BCG strains, as some BCG strains have RD2 deletion and some do not⁶. In our study we examined if baseline antibodies that recognize Mtb are associated with protection against infection, and based on our findings we can conclude this is not the case. Furthermore, BCG vaccination in the Dutch population (without any prior BCG vaccination and

small to none of Mtb exposure in the environment) only marginally increased levels of antibodies that recognize Mtb.

I also suggest harmonising language regarding these results, which are currently conflicting. In the results, lines 222-224 states "several Mtb specific IgG3 showed a statistically significant, albeit minimal increase, while several Mtb-specific IgM antibodies showed a minimal decrease. The discussion states in lines 240-241 "in a low incidence setting, adult BCG vaccination induced heterologous cytokine production, but did not lead to significant changes in anti-Mtb antibodies; in lines 321-324, "intradermal BCG vaccination of adults in a low-incidence setting..... did not significantly alter titers of Mtb-specific antibodies." The abstract states, " BCG vaccination induced heterologous innate cytokine production, but only marginally affected Mtb-specific antibody profiles".

Agree. Done.

Minor comments:

L 80-82 - sentence doesn't make grammatical sense.

It is now grammatically correct.

L 93 - does 'respectively' come after 'years of age'?

Agree. Done.

L 143 - indicate 'Mtb lysate', as you have now indicated in the methods.

Done

L 242-245 - these abbreviations should have been in the intro already

Done

References

1. Prendergast, K. A. *et al.* The Ag85B protein of the BCG vaccine facilitates macrophage uptake but is dispensable for protection against aerosol *Mycobacterium tuberculosis* infection. *Vaccine* **34**, 2608–2615 (2016).
2. Gonzalo-Asensio, J., Marinova, D., Martin, C. & Aguilo, N. MTBVAC: Attenuating the Human Pathogen of Tuberculosis (TB) Toward a Promising Vaccine against the TB Epidemic. *Front. Immunol.* **8**, (2017).
3. Cunningham, A. F. & Spreadbury, C. L. Mycobacterial Stationary Phase Induced by Low Oxygen Tension: Cell Wall Thickening and Localization of the 16-Kilodalton α -Crystallin Homolog. *J. Bacteriol.* **180**, 801–808 (1998).
4. Shi, C. *et al.* Enhanced protection against tuberculosis by vaccination with recombinant BCG over-expressing HspX protein. *Vaccine* **28**, 5237–5244 (2010).
5. Prinzis, S., Chatterjee, D. & Brennan, P. J. Structure and antigenicity of lipoarabinomannan from *Mycobacterium bovis* BCG. *Microbiology* **139**, 2649–2658 (1993).
6. Mahairas, G. G., Sabo, P. J., Hickey, M. J., Singh, D. C. & Stover, C. K. Molecular analysis of genetic differences between *Mycobacterium bovis* BCG and virulent *M. bovis*. *J. Bacteriol.* **178**, 1274–1282 (1996).